# Factors influencing fans' spectating experience and configuration effects in CBA league

Duan Yu [1], Hongwei Fan[2], Ning Zhang[1]*

1 Institute of Sport Training, Chengdu Sport University, Chengdu, China, 2 School of Economics and Law, Southwest University of Political Science and Law, Chongqing, China

* 417997356@qq.com

## Abstract

The low satisfaction of fans' spectator experience, the weak willingness to continue to watch games have become the realistic barriers limiting the sustained and favorable development of Chinese Basketball Association league (CBA League). Using Structural Equation Modeling (SEM) and Fuzzy-set Qualitative Comparative Analysis (fsQCA), this study clarifies the complex causal relationship behind the phenomenon of fans' spectator experience, explore the linkage mechanism between different influencing factors of spectator experience. Also constructs a variety of grouping paths to enhance the fans' spectator experience in the CBA League. The results shows that Sense, Feel, Think, Act, Relate and Spectator Service all positively affect the satisfaction of the live Spectator experience in the CBA League. Among six influencing factors, none of them is the necessary condition for high satisfaction of Fans' Spectator Experience. In this way, the characteristics of multi-factors and different paths of the live Spectator experience were verified. Influential factors were coupled to form six grouping paths for high satisfaction of the fans' spectator experience, which were categorized into two types of configurations by combining the grouping characteristics: 1) Led by spectator service, feel and act experience co-driven, 2) Spectator service and think experience co-driven. The findings provide theoretical guidance for enhancing fan experience and meeting the practical needs of cultivating loyal CBA fan groups while expanding the league's influence.

## 1 Introduction

As the highest-level men's basketball professional league in China, the China Basketball Association League (CBA League) has become a professional league with high influence in China after nearly 30 years of development, and also shoulders the mission and task of reserving and delivering basketball reserve talents for the country [1]. However, the low loyalty of the fan base [2], low consumption desire [3], and low attendance rate of the live games still inhibit the rapid development of the CBA league and the enhancement of its profitability [4], and it has become an urgent problem to improve the experience of the league's fans in the live games in

**Competing interests:** The authors have declared that no competing interests exist.

order to increase the number of people who watch the games on-site, thus promoting the rapid growth of the profitability of the CBA league. Whether it is the regular season, playoffs or all-star games, the number of on-site spectators is low, and the profitability of clubs is weak. The reason for this involves a variety of factors, including the level of the game, the operation of the game, the experience of watching the game and brand building [5]. From the perspective of practical implication, enhancing fans' on-site viewing experience lays the foundation for cultivating a loyal fan base, expanding the attractiveness of the league, and stimulating the development of the league's derivative economy [6]. In the reality that the profitability of CBA league is still in the doldrums, it is of high theoretical value and practical significance to explore the enhancement of on-site viewing experience from both theoretical and practical perspectives. According to this logic, it is urgent and important to explore the role mechanism and organizational path of the factors influencing the fans' experience of watching games in CBA league.

The innovations of this study are mainly reflected in: (1) through a large sample of empirical analysis, the SEM method is used to comprehensively verify the positive effects of Sense, Feel, Think, Act, Relate and Spectator Service quality on the live Spectator Experience of the CBA league, providing empirical data support for the study of the influencing factors of the live spectator experience. (2) The fsQCA method was innovatively applied to study the live spectator experience, and multiple factors were included in the same analytical framework to explore the synergistic and interactive effects of multiple factors on the live spectator experience from a group perspective, and different group paths were proposed to realize a highly satisfying live game spectator experience. (3) Using the empirical analysis method combining quantitative and qualitative methods, we accurately identify the core and marginal factors affecting the spectator of watching live games in CBA League. It clarifies the complex causal relationship behind the phenomenon of live spectator experience and explores the linkage mechanism between different influencing factors of on-site spectator experience.

## 2 Literature review

### 2.1 Factors influencing spectator experience

Sports events are essentially an "experiential" product for the audience, which is the "hard core" of the services provided by the entire stadium, and is also the core link of the industrial value chain of the CBA league. In Foroughi's study on fans' satisfaction with the live spectator experience, it is argued that ancillary services (entertainment of the event, arena factors, electronic facilities, and on-site hosts) as stimuli are the key factors influencing the generation of positive emotions, and in turn, affecting the live spectator experience of the fans[7]. Chang argues that the main intentions of fans who purchase a ticket to attend a live match are to seek pleasure or emotional fulfillment in the service experience, revealing that entertainment is an important motivation for fans to watch live [8]. In a study by Oztas and Bryan, it was found that large screens, high-quality sound systems and lighting, and visual factors play a crucial role in fans' live spectator experience as well as service quality evaluation. When venues have better electronic facilities, spectator satisfaction as well as the desire to stay is increased [9, 10]. In terms of the influencing factors of the spectator experience in the CBA League, the fans have the best perception of "sensory experience", followed by "emotional experience" and "thinking experience", indicating that the enhancement of sensory experience and service quality will play an important role in the evaluation of the quality of the service. "This suggests that strengthening the sensory and emotional experience strategy may be a more efficient means of resource utilization for the CBA management team [11]. After studying the main influencing factors of fans' spectating in China's professional soccer league, Li found that the

game level factor, the service level factor, the personal family factor, and the venue conditions factor constitute a multidimensional structural influencing factor system of fans' spectating in China's soccer league[12]. According to Tan, the main factors affecting fans' attendance are the low level of matches, the poor performance of teams in the league, the phenomena of "match-fixing" and "black whistles", etc. The only way to improve the quality of league matches is to continuously improve the quality of the league matches, to adopt appropriate marketing strategies, to establish brand awareness, and to cultivate a loyal fan base [13]. Only by continuously improving the quality of league matches, adopting appropriate marketing strategies, building brand awareness, and cultivating a loyal fan base can the satisfaction of fans and spectators be improved. According to Li, the attitude, subjective norms, and perceptual behavioral control have a positive predictive effect on the intention to watch the game, and the subjective norms have the strongest predictive effect [14]. In the era of prevalent experience economy, a good spectating experience is equally important for the cultivation of a loyal fan base. Fans' satisfaction after spectating a specific sports event is not only affected by a single variable, they may be affected by multiple variables such as expectation gap, sports team identification, opponent quality, empathy and viewing pleasure, etc. In addition, the strength of the emotional experience does not have a significant role in the formation of basketball spectators' satisfaction, but it has a direct impact on the basketball fans' loyalty to the team, i.e., the stronger the effect of the emotional experience, the higher the fan's loyalty [15].

## 2.2 Spectator service quality and satisfaction

Spectator service quality is mainly composed of outcome quality and functional quality, with outcome quality describing what is perceived by the customer after production and consumption [16]. Functional quality, on the other hand, is mainly related to service quality, including the interaction between service personnel and fans, the service items and service level provided, fan care, ease of access and transportation to and from the arena, and the safety of the arena [17]. At the end of the spectating, in addition to the on-site viewing atmosphere and physical sensations, the fans' evaluation of the spectating experience is still mainly determined by the quality of spectator services perceived during the viewing process [18]. Customer satisfaction with a service or product not only creates long-term benefits for an organization, but also is an important prerequisite for customer loyalty, repeat purchase intention, and the formation of positive word-of-mouth [19]. In studying the influencing factors of live spectator experience in the CBA league, the exploration of fan satisfaction is extremely necessary for both fostering a loyal fan base and cultivating positive behavioral intentions among fans.

In studies on spectator service quality, scholars generally agree that spectator service quality is a complex functional body composed of multiple service attributes. Murray points out that overall service quality is an important prerequisite for the formation of fans' perceived value [20]. Whereas Byon argued that the perceived value of fans is only influenced by home team factors and venue factors [21]. In addition, Yoshida stated that the attitude of the staff in the arena is crucial for enhancing satisfaction with the quality of spectator services [22]. On the contrary, Biscaia argued that fans' satisfaction with service quality is not only determined by the performance of the staff, but other factors such as the comfort and convenience of the arena also affect fans' satisfaction with service quality [23]. Lehtinen argued that service quality is mainly composed of three dimensions, namely, physical quality, corporate quality, and interactive quality, wherein Physical quality is the customer's perception of the quality of facilities and equipment during the service process, corporate quality is the customer's perceived corporate image and characteristics, and interaction quality is the interactive behavior between the customer group and the service personnel [24]. According to Rust and Oliver, in addition

to functional and technical quality, the service environment is also an important construct of service quality [25, 26]. In practical research, even though different scholars have different opinions on the concept of service quality, they all agree that service quality is an important driving factor to enhance customer perceptions of brand value, and that it is crucial to ensure service quality, whether it is the cultivation of a loyal relationship between the enterprise and its customer base or the realization of the enterprise's profitability.

Satisfaction, as an evaluation result produced by fans in the consumption process after comparing the preconceived ideas with the actual feelings, greatly affects the consumption intention of live fans [27]. After using structural equation modeling to construct a model of the influence of the viewing experience on the consumption intention of on-site spectators in the Chinese Super League, it is found that the viewing experience not only directly affects the consumption intention of on-site spectators, but also indirectly affects the consumption intention of on-site spectators through satisfaction [28]. The reason for this is that the viewing experience, as a result of the interaction between the spectators and the entire match, is mostly related to the team performance, the environment of the match, and the commentary on current affairs, etc. A good sensory experience can not only bring visual, auditory and other sensory system pleasure and enjoyment, but also stimulate the hedonic motivation of the spectators' desire to be entertained [29]. In terms of viewers' behavioral intention, Luo found that the spectator experience has a direct positive effect on spectator repurchase intention and recommendation intention after using PLS-SEM multivariate statistical method to test spectators' behavioral intention [30]. Once the fans form a sustained willingness to watch the game, it means that the psychological connection between the fans and the league has formed a high degree of resistance to change, and the behavioral intention is internally driven and shows self-identification [31].

After systematically combing through the relevant research literature, we found that there are still the following problems in the research on spectator experience. Existing research focuses on the impact of the spectator experience on fans' consumption behavior, fan loyalty and willingness to continue to watch the game, and few studies have explored the antecedent factors affecting the live spectator experience. Most of the studies on the spectator experience and the willingness to continue to watch the game use structural equation modeling, hierarchical regression analysis, factor regression analysis and mediation model, which can only explore the linear correlation between the variables, while the spectator experience of watching the game is the result of the compounding of multiple factors, and different combinations of the influencing factors will lead to different effects. The conclusion of the study needs to be enriched. The formation of spectator experience is a complex process, and there is no single necessary condition that can influence spectator experience. Exploring the multiple mechanisms of the influencing factors of fans' viewing experience will help enrich the conclusions of the current research on the viewing experience.

Using analysis methods of Structural Equation Modeling (SEM) and Fuzzy set Qualitative Comparative Analysis (fsQCA), this paper explores the influencing factors of the spectator experience in the CBA league and its enhancement paths based on the group perspective from both theoretical and empirical perspectives. Providing theoretical guidance and ideas for the enhancement of spectator experience, realize the practical needs of cultivating loyal fan groups of CBA League and expanding the influence of CBA itself.

## 3 Theoretical basis and research hypothesis

### 3.1 Experience economy theory and experiential marketing

The so-called experience economy refers to the fact that enterprises use services as the stage, goods as the props, and consumers as the center to create activities that can make consumers

participate and are worth remembering [32]. Tracing back to the theory of experience economy, the concept of "experience" was first proposed in the 1980s. Subsequently, Pine and Gilmore and others proposed the concept of customer experience and the experience economy as a new paradigm for marketing [33].

Experiential marketing as a marketing concept, its core value emphasizes that in the process of consumer services, to provide customers with impressive, emotional and unique positive experience, so as to cultivate the consumer's fascination with products and services [34]. In the process of identifying customer needs, meeting customer needs, and forming customer consumption behavior, companies establish good two-way communication and interaction channels with customers, and integrate corporate brand personality and value into customers' lives, so as to cultivate loyal customers and achieve the purpose of profitability [35]. Customers in the consumer experience felt in the service and the formation of emotional resonance with the corporate brand is also an important basis for customer behavioral loyalty and attitudinal loyalty [36].

In the conception and study of experiential marketing, the customer is no longer a purely rational consumer behavior decision maker, but a rational and emotional coexistence of the desire to realize personal fantasies, the pursuit of sensation and pleasure in the consumption experience of the "emotional animal" [37]. Compared with the traditional marketing approach in which marketers and consumers emphasize the functional attributes and value attributes of products, experiential marketing pays more attention to the experience of customers in the consumption process [38]. Considering the consumption process as an overall experience process, and expects to create a consumption scene that perfectly meets the psychological expectations of consumers, and at the same time provide customers with sensory, emotional, cognitive, behavioral, and relational values, thus weakening the functional and value attributes of products.

## 3.2 factors influencing the spectator experience based on the strategic experience module

Strategic Experiential Modules (SEMs) is a kind of experiential marketing strategy proposed by the famous American customer experience management expert Schmitt in 1999 based on the reflection on the traditional marketing method [39]. The core components of SEMs are Sense, Feel, Think, Action and Relate.

Sense refers to the process of consumer experience, through the stimulation of the visual, auditory, tactile, olfactory, taste and other sensory systems and the joint effect, so that the customer has a better sense experience. According to Kotler, the individual's psychological feelings, thoughts, attitudes and motivation will have an impact on the customer's purchasing behavior [40]. However, sometimes the stimuli received by the sensory system are not enough to generate consumer behavior, so the sensory experience needs to be combined with specific consumption scenarios to create unforgettable sensory experiences and memories for customers. In addition, Yoon showed that the quality of the product, the hardware, and the atmosphere of the experience positively affect the emotional response of the customer [41]. Similarly, in the on-site spectating experience, the hardware facilities of the CBA league venues, the atmosphere created, and the quality of the game as the core product are all important factors that fans can intuitively perceive and have an impact on the viewing experience. Therefore, the research hypothesis H1a on spectator experience is proposed.

H1a: The sense experience positively affects the fans' spectating experience.

Feel experience refers to the marketing strategy to stimulate the customer's inner feelings and emotions, so as to create a positive for the customer, to the for the company or brand to

feel pleasure, pride in the emotional experience. The intensity of the emotional experience formed in the process of contact and interaction increases over time. According to Wiggins, when a company is able to provide customers with a positive emotional experience on an ongoing basis, customers can ignore the negative aspects of the company or product to a certain extent and continue to be loyal to the brand [42]. The essence of fans' emotional experience in the process of watching live matches, whether it is the specific emotion of excitement or depression that arises with the match, or the quality of live service and the affectionate feeling towards the service staff that fans feel, all belong to the emotional experience of fans in the process of watching matches. Therefore, hypothesis H1b of the study on spectator experience is proposed.

H1b: Feel experience positively influences fans' spectating experience.

Think experience mainly refers to the thinking and intelligence activities carried out by consumers during the experience process, and its purpose in the experiential marketing strategy is to create cognitively meaningful problem-solving experiences for customers. By creating surprises, hints, and triggering customers to think, customers' thinking experience can be formed. For example, in the experience of visiting a science and technology museum, visitors' surprise, curiosity and inspiration are stimulated by the exhibits and the suggestive questions during the tour [43]. In the case of watching a live game, the thinking experience of the fans is reflected in the curiosity and thinking about the outcome of the game, as well as a series of thinking activities generated in the process of watching the game [44]. When the thinking experience is formed, the customer's connection with the enterprise or brand no longer relies solely on perceptual judgment, and the customer's thinking experience is not based on emotional judgment, but on the customer's own perception of the brand. is no longer purely based on emotional judgment, but on rational judgment based on in-depth thinking and evaluation. Accordingly, research hypothesis H1c on the spectator experience is proposed.

H1c: Think experience positively affects the spectator experience.

Act experience mainly refers to the concepts that customers feel in the process of consumption experience and convert the concepts into concrete behavior of actual action. In the process of conveying the concept, the enterprise also conveys different lifestyles and behavioral patterns to the customer, stimulating the customer to think about their own actual lifestyles. When customers convert the concept into practical actions, their recognition of the brand will be increased [45]. During the on-site viewing process, the promotional design and slogans of the CBA league venues will have a subtle influence on the fans. For example, the slogans "In the name of the city, stand for the glory of the city", "Basketball makes life better", will inspire the fans to change their concepts and actions, thus expressing their support for the CBA league. Thus, they express their support for the CBA league. Accordingly, research hypothesis H1d is proposed.

H1d: Act experience positively affects the spectating experience.

As a collection of sense experience, feel experience, think experience, and act experience, relate experience emphasizes the group relationship with other individuals beyond the feeling of self, and also reflects the individual's pursuit of group belonging and role modeling. Some scholars believe that when there are interactions and sharing behaviors in a specific area, individuals will develop a sense of dependence and belonging, and when the experience of association occurs, the connection between individuals will be closer, and the social relationship between individuals will be extended [46]. According to Lassar, the social image of a brand is essentially the customer's perception of the attitudes of its user group, which implicitly implies

that the customer's perception of the brand's functional properties and the expected positioning of the brand's users. and the expected positioning of the brand's users [47]. In live spectating process, the influence of associated experience on fans' live viewing experience is mainly reflected in the fact that when fans recognize and accept other fans, the group behavior effect arises, and fans' pursuit of group belonging, and dependence can be satisfied. Accordingly, research hypothesis H1e is proposed.

H1e: Relate experience positively influences the spectator experience.

### 3.3 factors influencing the spectator experience based on the spectator service quality

In spectator sports leagues where competition performance is the core product, satisfaction with the spectating experience has been shown to be an important factor in fostering a loyal fan base and increasing team revenues, as well as an important prerequisite for the formation of fans' willingness to continue to attend games [48]. The quality of spectator services at sporting events is mainly comprised of various programs as well as fans' perceptions of quality during the service delivery process [49]. In Carlson's study, it was found that fans' positive perceptions of service quality can significantly enhance the satisfaction of fans' spectating experience, which in turn fosters positive consumer behavioral intentions [50]. In addition, Yong found that all dimensions of spectator service quality (staff-spectator interactions, stadium hardware facilities, viewing atmosphere, and game results) significantly affect fans' overall service quality perceptions and indirectly influence behavioral intentions, including fans' satisfaction with the spectating experience, their willingness to continue to watch, and their purchasing behaviors [51]. Indirectly, it also affects the behavioral intention including fans' satisfaction with the spectating experience, their willingness to continue to watch the game and their purchasing behavior. Accordingly, research hypothesis H2a regarding the spectator experience is proposed.

H2a: Spectator service quality positively affects the spectator experience.

Based on the theoretical analysis conducted, the research hypotheses on the factors influencing the spectator experience constructed in this study are shown in Fig 1.

## 4 Methodology

### 4.1 Ethics statement

The study was conducted with the approval of the Ethics Review Committee of the Chengdu Sport University (No. 2024106) to which the principal investigator belongs. Written consent was obtained from the participants prior to the start of the study, and all participants signed a study-related informed consent form and chose to voluntarily participate in the questionnaire. If the participants were aged <18 years, consent was obtained from the participants and their guardians. In addition, a certificate of exemption from ethical review issued by the Ethical Review Committee of Chengdu Institute of Physical Education was obtained for this study.

### 4.2 Questionnaire design

Using the theoretical perspective method and the literature induction method, the Likert five-level scale of this study was constructed based on the Strategic Experiential module. The specific questionnaire is attached in S1 Appendix.

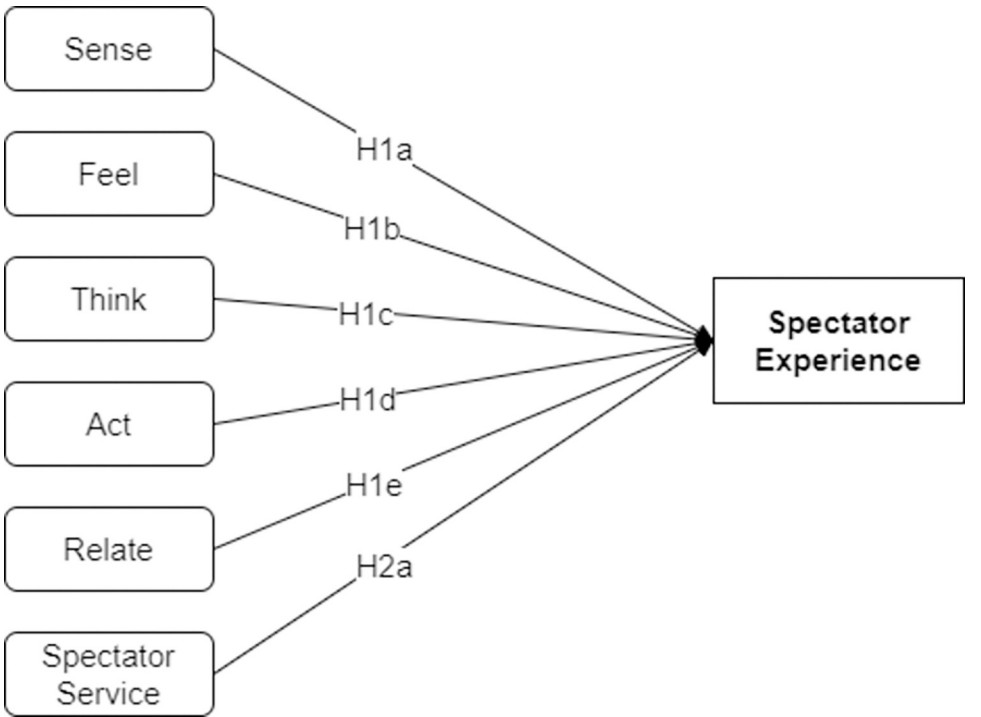

**Fig 1. Research hypotheses on factors influencing the spectator experience.**

The conditional variables consisted of Sense Experience, Feel Experience, Think Experience, Act Experience, Relate Experience [52], and Spectator Service [53]. The outcome variable then consisted of satisfaction with the spectating experience in terms of willingness to repurchase [54], future attendance [55], and tournament recommendations [56].

## 4.3 Data collection

The distribution of questionnaires mainly adopts the intentional sampling method and combines online and offline methods simultaneously. And questionnaires were distributed from April 15, 2023 to August 25, 2023. During the distribution process, in order to expand the interviewed group as much as possible, a total of 300 questionnaires were distributed through online links distributed to the fan groups of each home team in the CBA League, and offline face-to-face interviews. Discounting the questionnaires with inconsistent answers, consistent answer choices and those who have not had the experience of watching the game on site, a total of 274 valid questionnaires were recovered, and the recovery rate of valid questionnaires was 91.3%.

## 4.4 Descriptive statistical analysis of valid samples

After collating and counting the valid questionnaires, SPSS 26.0 was used to carry out descriptive statistical analysis of the valid samples, as shown in Table 1. The spectator who watched the games on the spot in the CBA League showed the gradual enlargement of the female fan group. In terms of age, the fan base of the CBA League reflects the youthfulness of the fan base of the CBA League, which also provides a reference for the practitioners of CBA League marketing in the formulation of marketing strategies. The times of on-site viewings indicates that the CBA League still has the problems of low fan conversion rate, weak will of the fans to

**Table 1. Descriptive statistical analysis of valid samples.**

| Variables | Categories | Frequency | Percentage |
|---|---|---|---|
| Sex | Male | 185 | 67.5% |
| | Female | 89 | 32.5% |
| Age | <18 | 10 | 3.7% |
| | 18–25 | 173 | 63.1% |
| | 26–30 | 70 | 25.5% |
| | 31–40 | 15 | 5.5% |
| | 41–50 | 6 | 2.2% |
| Education | Middle school | 4 | 1.5% |
| | High school | 13 | 4.7% |
| | specialized school | 45 | 16.4% |
| | Bachelor | 127 | 46.4% |
| | Postgraduate | 85 | 31.0% |
| times of on-site viewings | 1–2 | 108 | 39.3% |
| | 3–4 | 62 | 22.7% |
| | 5–6 | 26 | 9.5% |
| | >10 | 78 | 28.5% |

continuing watching, and loyalty to be improved. The emergence of the above problems may be related to the satisfaction of the on-site viewing experience, thus highlighting the practical value of this study. At the same time, the large proportion of 1–2 times on-site viewing also reflects that the CBA League has a strong attraction to fans and has a sizable potential fan base.

## 5 Analysis of influential factors on live spectator experience based on SEM

Structural Equation Modeling (SEM) is a method for building, estimating, and testing causal models. By replacing analytical methods such as multiple regression, path relationship analysis, factor analysis, and analysis of covariance, structural equation modeling is able to analyze the role of individual variables on the aggregate and the interrelationships between variables more clearly [57]. The structural equation modeling can more clearly analyze the role of individual variables on the whole and the interrelationship between variables. It can also better clarify the paths of the influencing factors on the outcome variables [58]. The measurement model is mainly composed of the observed variables, i.e., the Sense experience, the Feel experience, the Think experience, the Act experience, the Relate experience, and the Spectator Service, and the structural model is mainly composed of the relationship between the observed variables and the spectator experience.

### 5.1 Reliability and validity tests

As shown in Table 2. The Cronbach' α coefficients of the variables were 0.807, 0.847, 0.799, 0.815, 0.793, 0.799, 0.853, and 0.879 which were all between 0.7 and 0.8. It indicates that the scale in this study has good reliability. The CR value coefficients are all over 0.8, suggesting that the reliability of the scale is good.

Usually, the test of structural validity is mainly conducted through factor analysis, and before conducting factor analysis, KMO and Bartlett sphericity test are also essential, and factor analysis can be conducted only when the KMO value is greater than 0.5 and the p-value of Bartlett sphericity test is less than 0.05. As shown in Table 3. The KMO value is 0.904, which is

**Table 2. Results of scale reliability and validity tests.**

| Variables | Experience | Service | Relate | Act | Think | Feel | Sense |
|---|---|---|---|---|---|---|---|
| Experience | 0.512 | | | | | | |
| Service | 0.722** | 0.53 | | | | | |
| Relate | 0.585** | 0.394** | 0.54 | | | | |
| Act | 0.558** | 0.457** | 0.430** | 0.517 | | | |
| Think | 0.582** | 0.440** | 0.388** | 0.327** | 0.502 | | |
| Feel | 0.633** | 0.638** | 0.427** | 0.392** | 0.496** | 0.543 | |
| Sense | 0.442** | 0.340** | 0.359** | 0.304** | 0.298** | 0.388** | 0.550 |
| Cronbach' α | 0.807 | 0.847 | 0.815 | 0.793 | 0.799 | 0.853 | 0.879 |
| CR | 0.811 | 0.850 | 0.819 | 0.796 | 0.801 | 0.853 | 0.879 |
| AVE | 0.716 | 0.728 | 0.735 | 0.719 | 0.709 | 0.737 | 0.742 |

greater than 0.5, and the Bartlett's test of sphericity is 0.000, which is less than 0.05, indicating that the sample data of this study can be analyzed in the next step of factor analysis.

Shows in Table 4. The Chi-square/df is 1.100, which is less than 3, the root mean square of the error of approximation (RMSEA) is 0.019, which is less than 0.08, the goodness-of-fit index (GFI) is 0.901, the TLI is 0.987, and the comparative fit index CFI is 0.988, incremental modification index IFI is 0.988, and root mean square residual RMR is 0.036. The above values indicate that the model of influencing factors of spectator experience in the CBA league has a better degree of fit, and the structural validity of the scale is better.

## 5.2 Common method deviation test

Since the respondents who filled out the questionnaire were all from the fan base of the CBA league, a single source of data may cause the results of the study to be affected by common method bias. Therefore, all the scale items were put into a single factor of the spectator experience, and a one-way validation factor model was constructed to test the common method bias. If the fit indicators of the one-way validated factorial model were all above the test standard, and there was a large difference from the fit indicators of the original model, then it was considered that there was no problem of common method bias. As shown in Table 5. The chi-square/df of the common method bias test model is 4.324, which is greater than 3, the root mean square of the approximation error is 0.110, which is greater than 0.08, the goodness-of-fit index is 0.611, which is less than 0.9, the TLI index is 0.552, which is less than 0.9, the comparative fit index CFI is 0.581, which is less than 0.9, the incremental correction index IFI is 0.585, which is less than 0.9, and the root mean square residual RMR was 0.083, which is greater than 0.05. In summary, there is no common method bias in this study.

## 5.3 Structural equation model fitness and hypothesis testing

On the basis of a good degree of model fit, AMOS 24.0 was used to test the structural equation model (as shown in Fig 2) of the CBA league spectators' experience of watching live games.

**Table 3. KMO and Bartlett sphericity test values.**

| | | |
|---|---|---|
| KMO Measure of Sampling Adequacy | | 0.904 |
| Bartlett's Test of Sphericity | Approx. Chi-Square | 4000.780 |
| | df | 496 |
| | sig | 0.000 |

**Table 4. Confirmatory factor analysis model fit coefficients.**

| Indices | $X^2$/df | RMSEA | GFI | TLI | CFI | IFI | RMR |
|---------|----------|-------|-----|-----|-----|-----|-----|
| Criteria | <3 | <0.08 | >0.9 | >0.9 | >0.9 | >0.9 | <0.05 |
| Result | 1.100 | 0.019 | 0.901 | 0.987 | 0.988 | 0.988 | 0.036 |

The model fit test is to verify the degree of fit between the research data and the model, and indicators such as chi-square/degrees of freedom, RMSEA, GFI, TLI, CFI, IFI, RMR, etc., are commonly used to judge the overall degree of model fit.

As shown in Table 6, each test index of the structural equation model of the CBA league spectators experience meets the test standard. The degree of freedom of chi-square is 1.100, RMSEA is 0.019, GFI is 0.901, TLI is 0.987, CFI is 0.988, IFI is 0.988, and RMR is 0.036, and the results of the data analysis show that the structural equation model constructed in this study has a better fit.

As shown in Table 7. Sense Experience (β = 0.479, P<0.001), Feel Experience (β = 0.700, P<0.001), Think Experience (β = 0.608, P<0.001), Act Experience (β = 0.583, P<0.001), Relate Experience (β = 0.605, P<0.001), and Spectator Service (β = 0.753, P <0.001) all had a significant positive effect on spectator experience, and hypotheses H1a-H2a were supported by the data.

## 6 Analysis of the factors influencing the spectator experience based on fsQCA

As a specific consumption experience composed of multiple experience elements, the influencing factors are not independent, but interact with each other to form different degrees of spectator experience. Therefore, fuzzy-sets Qualitative Comparative Analysis (fsQCA) is further utilized to explore the mechanisms and linkages of the influencing factors of the spectator experience in the CBA league from a group perspective. Qualitative comparative analysis (QCA) is based on Boolean operations and set theory as a methodology, which focuses more on the configurational paths formed by multiple combinations of antecedent conditions on the outcome variable than traditional statistical tests that emphasize the "net effect" and have sufficient explanatory power [59]. The core conditions and edge conditions of the spectator experience are identified, and various grouping paths leading to better spectator experience are finally constructed, in order to realize the useful exploration of improving the quality of the live viewing experience of the fans in the CBA League. The theoretical analysis model of the influencing factors of the live viewing experience constructed in this study is shown in Fig 3.

**Table 5. One-way validated model test values.**

| Indices | One-way validation model | Original model fit | Criteria |
|---------|--------------------------|--------------------|----------|
| $X^2$/df | 4.324 | 1.100 | <1 |
| RMSEA | 0.110 | 0.019 | <0.08 |
| GFI | 0.611 | 0.901 | >0.9 |
| TLI | 0.552 | 0.987 | >0.9 |
| CFI | 0.581 | 0.988 | >0.9 |
| IFI | 0.585 | 0.988 | >0.9 |
| RMR | 0.083 | 0.036 | <0.05 |

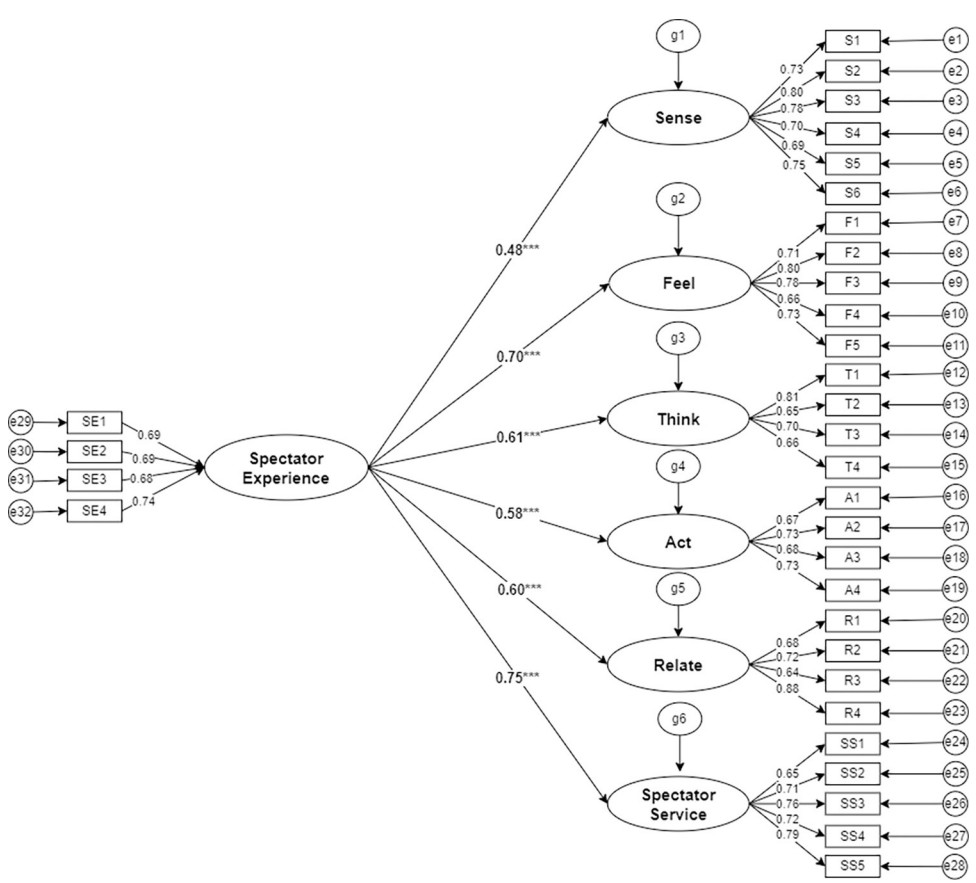

**Fig 2. Structural equation modeling of the factors influencing the spectator experience in CBA league.**

## 6.1 Variable selection and calibration

With reference to relevant studies and in conjunction with the distribution of the questionnaire data, the calibration method proposed by Ragin was followed to calibrate the raw data using the direct calibration method [60]. In terms of the setting of the affiliation anchor point, the mean of the condition variable and the outcome variable was taken as the intersection point. Based on this, the descriptive statistics of the condition variable and the outcome variable were analyzed using SPSS 26.0, and the respective standard deviations were used as the basis for affiliation anchoring. That is, the mean plus one standard deviation was considered fully affiliated and the mean minus one standard deviation was considered fully unaffiliated.

## 6.2 Necessity analysis

When the consistency level of the condition variable is between 0.8 and 0.9, it is regarded as a sufficient condition affecting the outcome variable, and when it is higher than 0.9, it is a

**Table 6. Structural equation modeling fitness tests.**

| Indices | $X^2$/df | RMSEA | GFI | TLI | CFI | IFI | RMR |
|---|---|---|---|---|---|---|---|
| Criteria | <3 | <0.08 | >0.9 | >0.9 | >0.9 | >0.9 | <0.05 |
| Model fits | 1.100 | 0.019 | 0.901 | 0.987 | 0.988 | 0.988 | 0.036 |

**Table 7. Research hypothesis testing results.**

| Path Correlation | Hypothesis | Path Coefficient | SE | CR | P | Result |
|---|---|---|---|---|---|---|
| Sense→Experience | H1a | 0.479 | 0.077 | 6.533 | *** | Supported |
| Feel→Experience | H1b | 0.700 | 0.080 | 8.729 | *** | Supported |
| Think→Experience | H1c | 0.608 | 0.093 | 8.101 | *** | Supported |
| Act→Experience | H1d | 0.583 | 0.082 | 7.105 | *** | Supported |
| Relate→Experience | H1e | 0.605 | 0.102 | 7.539 | *** | Supported |
| Service→Experience | H2a | 0.753 | 0.089 | 8.658 | *** | Supported |

necessity condition. The coverage value, on the other hand, reflects the strength of explanation of the outcome variable by the condition variable.

As shown in Table 8, the consistency level of each conditional variable of live viewing experience is below 0.8, and there is no single necessary condition that constitutes a high-quality live viewing experience. The absence of the necessary condition indicates that the live viewing experience of CBA league fans is the result of the complex effect of various influencing factors, so it is necessary to further analyze the role of the combination of conditional variables in enhancing the live viewing experience.

In contrast to the QCA method, the NCA method is able to quantitatively analyze the extent to which the antecedent conditioning variables need to be met in order to achieve a high level of satisfaction with the live viewing experience. Two methods, regression ceiling analysis (CR) and envelope ceiling analysis (CE), were used to calculate the effect size values of the six antecedent condition variables. In the analysis results, the effect size values were values between 0 and 1, with smaller effect size values indicating smaller effects. According to the recommendations of related research, the necessary conditions calculated by the NCA method must satisfy two conditions at the same time: (1) the effect value is not less than 0.1, and (2) the results of the Monte Carlo simulations of permutation tests (MCS) show that the effect sizes are significant, i.e., the P-value is greater than 0.5.

As shown in Table 9, the effect sizes of the six antecedent condition variables are all less than 0.1 and none of the effect sizes reach the significant level; therefore, among the six antecedent condition variables of the live spectator experience, a single condition does not constitute a necessary condition for a highly satisfying live spectator experience.

Bottleneck level analysis refers to the value of the level of the antecedent condition variable that is required to achieve a high level of satisfaction with the live viewing experience within

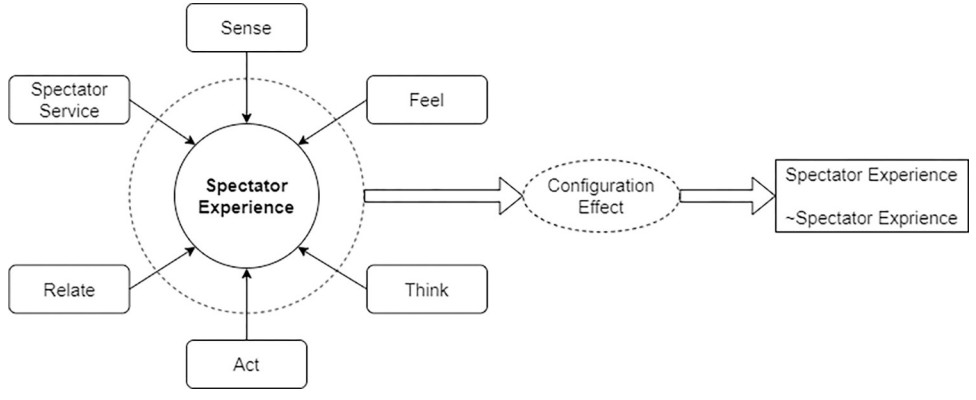

**Fig 3. Theoretical analysis model of the factors influencing the spectator experience based on fsQCA.**

**Table 8. Analysis of the necessary conditions for spectator experience.**

| Variables | Satisfied spectator experience | | ~Spectator experience | |
|---|---|---|---|---|
| | Consistency | Coverage | Consistency | Coverage |
| Sense | 0.751 | 0.755 | 0.648 | 0.563 |
| ~Sense | 0.566 | 0.651 | 0.718 | 0.713 |
| Feel | 0.754 | 0.770 | 0.606 | 0.534 |
| ~Feel | 0.545 | 0.616 | 0.740 | 0.721 |
| Think | 0.767 | 0.764 | 0.607 | 0.607 |
| ~Think | 0.521 | 0.606 | 0.726 | 0.729 |
| Act | 0.793 | 0.785 | 0.612 | 0.523 |
| ~Act | 0.518 | 0.608 | 0.748 | 0.757 |
| Relate | 0.783 | 0.763 | 0.655 | 0.551 |
| ~Relate | 0.539 | 0.644 | 0.719 | 0.741 |
| Spectator service | 0.797 | 0.807 | 0.565 | 0.494 |
| ~Spectator service | 0.500 | 0.571 | 0.779 | 0.768 |

the maximum observed range of the antecedent condition variable and the outcome condition variable. As shown in Table 10, to achieve a 100% level of satisfaction with the live viewing experience, it would be necessary to increase the level of the sensory experience by 5.2%, the affective experience by 19.5%, the thinking experience by 9%, the action experience by 11.7%, the associative experience by 3%, and the fan service level by 6.7%., 3.2% of the connected experience, and 6.4% of the fan service level.

## 6.3 Configuration sufficiency analysis

As the core of the QCA methodology, conditional grouping analysis aims to explore the effects that different groupings of antecedent conditions have on the outcome variables, and the process of exploring whether the set of grouping paths composed of multiple conditioning variables is a subset of the set of outcome variables.

**6.3.1 Configuration path for highly satisfying spectator experience.** The QCA method was used to analyze the conditional grouping of CBA league spectator experience. As shown in Table 11, the consistency levels of the six grouping paths are 0.939, 0.938, 0.946, 0.935, 0.937, and 0.945, respectively, which all satisfy the test of the sufficiency condition of more than 0.9.

**Table 9. NCA-based necessary condition analysis.**

| variables | Method | Accuracy% | ceiling zone | Scope | d | P |
|---|---|---|---|---|---|---|
| Sense | CR | 100% | 0.003 | 0.92 | 0.004 | 0.108 |
| | CE | 100% | 0.006 | 0.92 | 0.007 | 0.100 |
| Feel | CR | 97.4% | 0.042 | 0.92 | 0.045 | 0.000 |
| | CE | 100% | 0.025 | 0.92 | 0.027 | 0.000 |
| Think | CR | 99.6% | 0.026 | 0.90 | 0.029 | 0.000 |
| | CE | 100% | 0.030 | 0.90 | 0.033 | 0.003 |
| Act | CR | 98.9% | 0.011 | 0.90 | 0.013 | 0.010 |
| | CE | 100% | 0.013 | 0.90 | 0.015 | 0.019 |
| Relate | CR | 100% | 0.002 | 0.90 | 0.002 | 0.368 |
| | CE | 100% | 0.004 | 0.90 | 0.004 | 0.335 |
| Spectator service | CR | 99.3% | 0.020 | 0.90 | 0.022 | 0.005 |
| | CE | 100% | 0.021 | 0.90 | 0.023 | 0.027 |

**Table 10. Bottleneck level analysis of the spectator experience.**

|  | Sense | Feel | Think | Act | Relate | Service |
|---|---|---|---|---|---|---|
| 0 | NN | NN | NN | NN | NN | NN |
| 10 | NN | NN | NN | NN | NN | NN |
| 20 | NN | NN | NN | NN | NN | NN |
| 30 | NN | NN | NN | NN | NN | NN |
| 40 | NN | NN | 0.7 | NN | NN | 0.9 |
| 50 | NN | NN | 2.1 | NN | NN | 1.8 |
| 60 | NN | 2.7 | 3.5 | NN | NN | 2.7 |
| 70 | NN | 6.9 | 4.8 | NN | NN | 3.7 |
| 80 | NN | 11.1 | 6.2 | 0.8 | NN | 4.6 |
| 90 | 1.4 | 15.3 | 7.6 | 6.2 | 0.9 | 5.5 |
| 100 | 5.2 | 19.5 | 9.0 | 11.7 | 3.2 | 6.4 |

It better presents the characteristics of high-quality cases of CBA league fans watching the game on site. Combined with the grouping characteristics, the six grouping paths were categorized into two configurations: (1) Led by spectator service, feel and act experience co-driven (2) Spectator service and think experience co-driven.

1. *Led by spectator service, feel and act experience co-driven.* This configuration mainly covers two configuration paths, path 1a and path 1b. In the process of watching the game, the fans' profound feel experience and positive act experience, together with the high level and quality of spectator service in the venue are the main features of this configuration, forming a grouping of the game-watching experience in which the fans' feel experience and act experience play a joint role in the process of watching the game. In path 1a, Feel experience, Act experience and Spectator service exist as core conditions, and Sense experience exists as a marginal condition. It indicates that in the process of live viewing, regardless of the fans' think experience and relate experience, as long as the fans are given better feel experience and act experience, so that the fans feel a better level of on-site service, supplemented by the sense experience, the fans will experience a better live viewing experience. In terms of explanatory strength, the consistency of path 1a was 0.939, the original coverage was 0.498, and the unique coverage was 0.009. indicating that 49.8% of all cases studied could be

**Table 11. Configuration path for highly satisfying spectator experience.**

| Variables | Path 1a | Path 1b | Path 2a | Path 2b | Path 2c | Path 2d |
|---|---|---|---|---|---|---|
| Sense | • |  | • | • |  | ⊗ |
| Feel | ● | ● |  | • |  | • |
| Think |  |  | ● | ● | ● | ● |
| Act | ● | ● | • |  | • |  |
| Relate |  | • |  | ⊗ | • | • |
| Spectator service | ● | ● | ● | ● | ● | ● |
| Raw coverage | 0.498 | 0.509 | 0.480 | 0.318 | 0.491 | 0.343 |
| Unique coverage | 0.009 | 0.015 | 0.007 | 0.021 | 0.011 | 0.017 |
| Consistency | 0.939 | 0.946 | 0.938 | 0.937 | 0.935 | 0.945 |
| Solution coverage | 0.666 | | | | | |
| Solution consistency | 0.918 | | | | | |

Note: ● = core condition present; ⊗ = core condition missing; • = borderline condition present; ⊗ = borderline condition missing; blanks indicate may or may not be present

explained by this path.

In path 1b, Feel experience, Act experience and Spectator Service are the core conditions of the configuration, and Relate experience exists as a marginal condition. It means that in the process of watching the game, when the satisfaction of fans' sense experience and think experience is low, the feel experience of fans in the process of watching the game can be improved by guiding the outburst of fans' emotions and the generation of positive emotional feelings. At the same time, the interaction between fans and the game can be increased to create a good act experience. In addition, a better level of spectator service is also important to improve the overall satisfaction of the viewing experience. The role of relate experience is reflected in the satisfaction of the sense of belonging of the fan community. The consistency of Grouping 1b is 0.946, the original coverage is 0.509, and the unique coverage is 0.015. Among the six grouping paths, this grouping is able to explain the largest number of cases, reaching 50.9%. Unlike path 1a, relate Experience was present as an marginal condition in path 1b. It is suggested that the fulfillment of the collective sense of belonging and the sense of identification with others during the live viewing process enhances the spectator experience of homophobic fans. The most obvious symptom is that this type of fans tend to participate in collective actions such as cheering during the game, and take it as the main goal of the game, so as to seek social belonging and other people's identity. In practice, in the operation practice of the CBA league, fans should be guided to participate in positive collective behaviors, and the sense of collective belonging should be used as a medium to enhance the recognition and loyalty of fans to the brand and culture of the CBA league.

2. *Spectator service and think experience co-driven.* Spectator service and think experience co-driven mainly includes four paths of path 2a, 2b, 2c and 2d. The antecedent configuration of path 2a is "Sense experience * Think experience * Act experience * Spectator Service", in which Think experience and Spectator Service are the core conditions of this state, and Sense experience is the marginal condition of this path. This suggests that in the on-site viewing experience, when the feel experience and relate experience of the fan group are low, the fans can be stimulated to think and provide high-level on-site spectator services, so that the CBA League fans have a better on-site viewing experience. In addition, sense experience and act experience as marginal conditions show that the atmosphere created by the existing audio-visual effects of the CBA league venues and the interaction with the fan groups enhance the on-site viewing experience of the CBA league fans to a certain extent, but the enhancement effect is not significant. It suggests that the atmosphere of the CBA League needs to be improved and the interactive channels for fans need to be broadened. The consistency level of pattern 2a is 0.938, the original coverage is 0.480, and the unique coverage is 0.007, which means that 48% of all cases can be explained by this pattern.

The combination of conditions for path 2b was Sense Experience* Feel Experience * Think Experience * ~Relate Experience * Spectator Service. The core conditions are Think experience and Spectator Service, the marginal conditions are Sense experience and Feel experience, and Relate experience is missing as a marginal condition. Path 2b shows that in the live viewing experience of the CBA league, the fans' positive thinking activities and perfect spectator services during the live viewing process are the core elements to ensure that the fans of the CBA league have a better spectator experience. The better audio-visual effect of the arena and the emotion of the fans to the league play a role in improving the satisfaction of the on-site viewing experience, and the lack of the relate experience shows that the interaction between the CBA league and the fan groups is still lacking, and there is the problem of the lack of interactive channels. The consistency level of grouping 2b is 0.937, the original

coverage is 0.318, the unique coverage is 0.021, and this grouping is able to explain about 31.8% of all studied cases.

The conditional groupings of path 2c are Think Experience*Act Experience*Relate Experience*Spectator Service. This indicates that in the process of watching CBA games on-site, as long as it can stimulate the thinking activities of the fan community and provide better quality of on-site spectator services, a higher satisfaction of the spectator experience can be realized. The act experience and relate experience play an auxiliary role for the improvement of the satisfaction of the spectator experience. In the path 2c, the think experience and spectator service are the core conditions for the high satisfaction of spectator experience, and the act experience and relate experience are the auxiliary conditions. The consistency of this grouping is 0.935, the original coverage is 0.491, and the unique coverage is 0.011, which can explain about 49.1% of the research cases, and the consistency level suggests that there is a 93.5% probability of realizing the high satisfaction of the fans' live viewing experience after following this grouping path in the viewing experience of the CBA league.

The conditional grouping of path 2d is ~Sense Experience *Feel Experience * Think Experience *Relate Experience * Spectator Service. It means that the active thinking activities of fans and the high level of spectator service play an important role in realizing the high satisfaction of the spectator experience. Consistent with path 2a, 2b, and 2c, the core conditions of Group 2d were Think Experience and Spectator Service, and the marginal conditions were Feel Experience and Relate Experience. The absence of the sense experience condition variable indicates that in the current CBA league's on-site viewing experience, whether it is in terms of the arena or in terms of the game itself, the sense experience of the fans in the process of watching the game still needs to be improved. For example, the cleanliness and design of the arena, the audio-visual effect during the viewing process and the smoothness of the game. The marginal conditions of feel experience and relate experience indicate that the role of establishing the emotional connection between fan groups and the league, players and teams, as well as satisfying the fans' sense of belonging to the group so as to achieve a high degree of relate between the fans and the league can not be ignored in improving the satisfaction of the spectator experience of the CBA league. The consistency of path 2d is 0.945, the original coverage is 0.342, and the unique coverage is 0.017. Among the six grouping paths, the explanation of path 2d is weak, only explaining about 34.2% of the study cases.

## 6.4 Robustness test

The method of adjusting the PRI consistency and frequency threshold is adopted to test the robustness of the grouping path of high-satisfaction live viewing experience in CBA league, and to test whether the grouping is robust through the change of conditional grouping and related parameters. The specific operation process is to improve the PRI consistency from 0.8 to 0.9, adjust the frequency threshold from the initial 3 to 6 and then carry out the normalization operation again. The conditional grouping of the intermediate, parsimonious, and complex solutions after the operation is consistent with that of the three solutions before the adjustment, and the original coverage, unique coverage, and consistency level of individual solutions are unchanged, which are consistent with the original grouping path. Therefore, it is shown that the grouping results are robust.

# 7 Discussion and suggestion

## 7.1 Discussion

1. Structural equation modeling analysis shows that in the process of on-site viewing, Sense Experience, Feel Experience, Think Experience, Act Experience, Relate Experience and

Spectator Service positively affect the Spectator Experience of CBA League, i.e., the better the above experience is, the higher the satisfaction of spectator experience is. The path test coefficients are 0.479 for Sense Experience, 0.700 for Feel Experience, 0.608 for Think Experience, 0.583 for Act Experience, 0.605 for Relate Experience, and 0.753 for Spectator Service, and the hypotheses of H1a, H1b, H1c, H1d, H1e, and H2a of the study are all verified.

2. The results of the necessity condition analysis show that among the six influencing factors of the spectator experience in the CBA league, single influencing factor does not constitute a necessary condition for high-satisfaction spectator experience. That is to say, through a variety of combinations of different influencing factors, it is still possible to realize the high satisfaction of spectator experience. In this way, the characteristics of multi-factor concurrency and different paths to the same destination of the live game viewing experience were verified. The bottleneck level analysis shows that 5.2% of Sense Experience, 19.5% of Feel Experience, 9% of Think Experience, 11.7% of Act Experience, 3.2% of Relate Experience and 6.4% of Spectator Service are needed to improve to realize high satisfaction of Spectator Experience. This reflects that the emotional connection between fans and the CBA league is weak, and more humanistic care needs to be given to the fan base to improve their action experience and integrate the league concept into their own lives.

3. The analysis of grouping adequacy shows that there are six grouping paths to realize the high satisfaction spectator experience of CBA league fans. Combined with their grouping characteristics, were categorized into two configurations: (1) Led by spectator service, feel and act experience co-driven (2) Spectator service and think experience co-driven.

Among them, the spectator service condition variable is the core influence factor in the six grouping paths. It indicates that the CBA league should focus on the construction of the spectator service system to improve the service level and quality during the live viewing process. Regarding the degree of explanation of the grouping, the explanation of the grouping 2a with the condition combination of Feel Experience*Act Experience*Relate Experience*Spectator Service is the highest, reaching 50.9%, and the Feel Experience, Act Experience, and Spectator Service are the core conditional variables of path 2a.

In addition to internal influences such as game and service, some external factors may also affect the CBA league fans' on-site viewing experience. For example, perceptions of event prestige play an important role in influencing fans' willingness to continue to attend games, and sports event organizers can adopt different strategies to develop events and maintain fans' interest in basketball [61]. Sport experience and spectator type, on the other hand, mediate the emergence of event nostalgia, as spectators with the same sport experience tend to be more loyal to the event [62]. Additionally, regional cultural identities also influence the sensory experience of the fan base, and the effectiveness of the sensory-centered experience is enhanced when regional cultural identities and elements are integrated into the arena layout [63].

## 7.2 Suggestion

1. Emphasize the fans' demand and provide on-site viewing services that meet the fans' real demand. To enhance the effectiveness and pertinence of the CBA league competition and performance service supply, the establishment of the demand communication mechanism should start from the optimization of the top-level design, smooth the top-down demand communication channels, and retain the fans' right to express their demand and participate in the decision-making process.

2. Promote the intelligentization of the CBA league tournament governance environment system. Through the introduction of digital intelligent governance means and the establishment of tournament data system, real-time monitoring, effective management and timely adjustment of tournament activities, to realize the active governance of tournament activities.

3. Take the fans as the center, and build a fan service system with the concept of humanistic care. The construction of fan service system should realize the perfect fusion of tangible services and intangible feelings, and the perfect fan service system should not only be reflected in the game watching site, but also cover the community field outside the game. The public welfare activities and open training days are conducive to the emotional integration between the fans and the clubs, and realize the loyalty of the fans as well as the improvement of the brand culture identity of the league.

4. To create intelligent CBA league, technology helps to enhance the on-site viewing experience. Introduced Various kinds of intelligent equipment and electronic facilities to lay the material foundation and environmental conditions for the creation of in-depth interaction and immersive viewing atmosphere for the fans on site, with a view to continuously improving the on-site viewing experience of the fans in the CBA League.

## 7.3 Limitations and prospects

This study has the following limitations: (1) Although this study conducted a common method bias test for the questionnaire results, the effectiveness of the study could have been improved by using more diverse methods of data collection and triangulation, such as interviews, observations, and cross-validation of different indicators. (2) Due to the operationalization of variables and the difficulty of data collection, relevant external influences were not included in this study. Future studies may consider incorporating external factors such as regional economic development, public interest in basketball, and global sports development patterns.

## Supporting information

**S1 Appendix. Questionnaire.**
(DOCX)

**S2 Appendix. Data.**
(XLSX)

## Author Contributions

**Conceptualization:** Duan Yu, Hongwei Fan, Ning Zhang.

**Data curation:** Duan Yu.

**Formal analysis:** Duan Yu.

**Investigation:** Hongwei Fan.

**Methodology:** Duan Yu, Hongwei Fan.

**Software:** Duan Yu, Hongwei Fan.

**Supervision:** Ning Zhang.

**Validation:** Ning Zhang.

**Writing – original draft:** Duan Yu, Hongwei Fan.

**Writing – review & editing:** Duan Yu, Ning Zhang.

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
