## [Decision Letter · Decision Letter 0]

14 Nov 2024

PONE-D-24-35733Factors influencing fans' spectating experience and configuration effects in CBA leaguePLOS ONE

Dear Dr. Zhang,

Thank you for submitting your manuscript to PLOS ONE. After careful consideration, we feel that it has merit but does not fully meet PLOS ONE’s publication criteria as it currently stands. Therefore, we invite you to submit a revised version of the manuscript that addresses the points raised during the review process.

We look forward to receiving your revised manuscript.

Kind regards,

Vincenzo Auriemma

Academic Editor

PLOS ONE

**Journal Requirements:**

2. Please provide additional details regarding participant consent. In the ethics statement in the Methods and online submission information, please ensure that you have specified (a) whether consent was informed and (b) what type you obtained (for instance, written or verbal, and if verbal, how it was documented and witnessed). If your study included minors, state whether you obtained consent from parents or guardians. If the need for consent was waived by the ethics committee, please include this information.

**Additional Editor Comments:**

**Reviewer 1:**

The article titled Factors influencing fans' spectating experience and configuration effects in CBA League is interesting. I do not know the length for publication. This article appears really long, with over 50 pages. According to the journal criteria for publication, please see below my evaluation:

1. The study presents an original research. The authors use structural equation modeling (SEM) and fuzzy set qualitative comparative analysis (fsQCA) to analyze factors influencing fans' spectating experience in the CBA league, which is clearly an empirical research study based on newly collected data from 274 questionnaires.

2. I have searched online; there is no evidence to suggest that this work has been published elsewhere.

3. it seems to me that the analyses are conducted to a high technical standard. The study uses well-regarded methods such as SEM and fsQCA, with detailed descriptions of how these analyses were performed. The article also provides reliability and validity tests to ensure the robustness of the findings.

4. Overall, the article is presented in clear, standard English, with logical structuring, and appropriate technical language. The manuscript is well-organized and includes detailed sections such as introduction, literature review, methodology, results, and discussion. Some minor comments, please see below.

5. I think the conclusions are appropriately presented and supported by the data. The authors use multiple statistical methods, including SEM and fsQCA, to ensure that their conclusions are backed by robust data analysis. The research hypotheses are tested and confirmed using these methods, and the findings are clearly outlined with sufficient detail.

I would also like the authors to consider the following comments. Hope this could help the authors to improve the quality of the article:

1. Use of abbreviations: Some abbreviations, like CBA (Chinese Basketball Association), are used inconsistently with the full term, even after being introduced. For example, "CBA league" is mentioned in various forms (CBA, Chinese Basketball Association, etc.), which could be unified for better readability.

2. I would suggest to rewrite this sentence 'The findings of this study provide theoretical guidance and ideas for the enhancement of fans' spectator experience, and realize the practical needs of cultivating loyal fan groups of CBA league and expanding the influence of CBA itself.' as 'The findings provide theoretical guidance for enhancing fan experience and meeting the practical needs of cultivating loyal CBA fan groups while expanding the league's influence.'.

3. The sentence 'The analysis of necessity conditions shows that individual influencing factors do not constitute the necessary conditions for high satisfaction of Fans’ Spectator Experience.' and 'Starting from the group perspective, we use the empirical analysis method combining SEM and fsQCA to clarify the complex causal relationship' are not clear. I would recommend the authors to rewrite these two sentences.

4. The questionnaires results are not always reliable; the study would be stronger with a more diverse data collection with a triangulation methodology, such as interviews, observations, or cross-validation with ticket sales or fan engagement metrics. If the authors can not do this in this article, I would recommend the authors to put this as one of the limitation of this research.

5. I would also suggest the authors consider other external factors like broader economic conditions, public interest in basketball, and global sports trends are not considered in the study. These external influences might play a significant role in shaping fan experiences, but the research focuses only on internal factors related to the games and services.

**Reviewer 2:**

My opinions are for reference only. If there is a conflict with the editor's opinion, please follow the editor's opinion as the main one

1. The abstract is too redundant and more like a research conclusion. Suggest the author to simplify

2. There is a lack of relevant literature as a basis in the research background, and the author needs to search for relevant literature as a supplement

3.Provide a detailed description of the research methodology for fsqca, https://doi.org/10.3390/systems12020053

4.Suggest focusing on“ Experience Economy Theory and Experiential Marketing” conducts different configuration analyses

Reviewers' comments:

Reviewer's Responses to Questions

**Comments to the Author**

1. Is the manuscript technically sound, and do the data support the conclusions?

Reviewer #1: Yes

Reviewer #2: Partly

2. Has the statistical analysis been performed appropriately and rigorously? 

Reviewer #1: Yes

Reviewer #2: Yes

3. Have the authors made all data underlying the findings in their manuscript fully available?

Reviewer #1: No

Reviewer #2: Yes

4. Is the manuscript presented in an intelligible fashion and written in standard English?

Reviewer #1: Yes

Reviewer #2: No

5. Review Comments to the Author

**Reviewer #1: **The article titled Factors influencing fans' spectating experience and configuration effects in CBA League is interesting. I do not know the length for publication. This article appears really long, with over 50 pages. According to the journal criteria for publication, please see below my evaluation:

1. The study presents an original research. The authors use structural equation modeling (SEM) and fuzzy set qualitative comparative analysis (fsQCA) to analyze factors influencing fans' spectating experience in the CBA league, which is clearly an empirical research study based on newly collected data from 274 questionnaires.

2. I have searched online; there is no evidence to suggest that this work has been published elsewhere.

3. it seems to me that the analyses are conducted to a high technical standard. The study uses well-regarded methods such as SEM and fsQCA, with detailed descriptions of how these analyses were performed. The article also provides reliability and validity tests to ensure the robustness of the findings.

4. Overall, the article is presented in clear, standard English, with logical structuring, and appropriate technical language. The manuscript is well-organized and includes detailed sections such as introduction, literature review, methodology, results, and discussion. Some minor comments, please see below.

5. I think the conclusions are appropriately presented and supported by the data. The authors use multiple statistical methods, including SEM and fsQCA, to ensure that their conclusions are backed by robust data analysis. The research hypotheses are tested and confirmed using these methods, and the findings are clearly outlined with sufficient detail.

I would also like the authors to consider the following comments. Hope this could help the authors to improve the quality of the article:

1. Use of abbreviations: Some abbreviations, like CBA (Chinese Basketball Association), are used inconsistently with the full term, even after being introduced. For example, "CBA league" is mentioned in various forms (CBA, Chinese Basketball Association, etc.), which could be unified for better readability.

2. I would suggest to rewrite this sentence 'The findings of this study provide theoretical guidance and ideas for the enhancement of fans' spectator experience, and realize the practical needs of cultivating loyal fan groups of CBA league and expanding the influence of CBA itself.' as 'The findings provide theoretical guidance for enhancing fan experience and meeting the practical needs of cultivating loyal CBA fan groups while expanding the league's influence.'.

3. The sentence 'The analysis of necessity conditions shows that individual influencing factors do not constitute the necessary conditions for high satisfaction of Fans’ Spectator Experience.' and 'Starting from the group perspective, we use the empirical analysis method combining SEM and fsQCA to clarify the complex causal relationship' are not clear. I would recommend the authors to rewrite these two sentences.

4. The questionnaires results are not always reliable; the study would be stronger with a more diverse data collection with a triangulation methodology, such as interviews, observations, or cross-validation with ticket sales or fan engagement metrics. If the authors can not do this in this article, I would recommend the authors to put this as one of the limitation of this research.

5. I would also suggest the authors consider other external factors like broader economic conditions, public interest in basketball, and global sports trends are not considered in the study. These external influences might play a significant role in shaping fan experiences, but the research focuses only on internal factors related to the games and services.

**Reviewer #2: **My opinions are for reference only. If there is a conflict with the editor's opinion, please follow the editor's opinion as the main one

1. The abstract is too redundant and more like a research conclusion. Suggest the author to simplify

2. There is a lack of relevant literature as a basis in the research background, and the author needs to search for relevant literature as a supplement

3.Provide a detailed description of the research methodology for fsqca, https://doi.org/10.3390/systems12020053

4.Suggest focusing on“ Experience Economy Theory and Experiential Marketing” conducts different configuration analyses

6. PLOS authors have the option to publish the peer review history of their article (what does this mean?). If published, this will include your full peer review and any attached files.

Reviewer #1: No

Reviewer #2: No

---

## [Author Response · Author response to Decision Letter 0]

2 Dec 2024

Dear Editor Vincenzo Auriemma and Reviewers:

On behalf of my co-authors, we sincerely appreciate you and the reviewers for your precious time in reviewing our paper and giving us another chance to revise our manuscript. It was your valuable and insightful comments that led to possible improvements in the revised paper. According to you and the reviewers’ comments, we have carefully considered the comments and tried our best to address every one of them. We hope the manuscript after careful revisions can meet your high standards and expectations. Below we provide the point-by-point responses, all modifications in the manuscript have been highlighted in yellow.

Once again, we would like to express our great appreciation to you and the reviewers for your warm work and comments on our paper. We look forward to hearing from you!

Thank you and best regards,

---

## [Editor Report · Decision Letter 1]

16 Dec 2024

Factors influencing fans' spectating experience and configuration effects in CBA league

PONE-D-24-35733R1

Dear Dr. Zhang,

We’re pleased to inform you that your manuscript has been judged scientifically suitable for publication and will be formally accepted for publication once it meets all outstanding technical requirements.

Kind regards,

Vincenzo Auriemma

Academic Editor

PLOS ONE
---

## [Editor Report · Acceptance letter]

20 Dec 2024

PONE-D-24-35733R1 

PLOS ONE

Dear Dr. Zhang, 

I'm pleased to inform you that your manuscript has been deemed suitable for publication in PLOS ONE. Congratulations! Your manuscript is now being handed over to our production team.

Kind regards, 

on behalf of

Dr. Vincenzo Auriemma 

Academic Editor

PLOS ONE